# Endotyping of Cholesteatoma: Which Molecular Biomarkers? A Systematic Review

**DOI:** 10.3390/jpm12081347

**Published:** 2022-08-21

**Authors:** Elena Cantone, Claudio Di Nola, Eugenio De Corso, Michele Cavaliere, Giusi Grimaldi, Anna Rita Fetoni, Gaetano Motta

**Affiliations:** 1Department of Neuroscience, Reproductive and Odontostomatological Sciences-ENT Section, University of Naples Federico II, 80131 Naples, Italy; 2Head and Neck Department-ENT Section, AOU Federico II, 80131 Naples, Italy; 3Head and Neck Surgery-Otorhinolaryngology, Fondazione Policlinico Universitario A. Gemelli IRCCS, 00168 Rome, Italy; 4Department of Neuroscience, Reproductive and Odontostomatological Sciences-Audiology Section, University of Naples Federico II, 80131 Naples, Italy; 5Head and Neck Department-Audiology Section, AOU Federico II, 80131 Naples, Italy; 6Otorhinolaryngology, Head and Neck Surgery Unit, Department of Mental and Physical Health and Preventive Medicine, Università degli Studi della Campania Luigi Vanvitelli, 80131 Naples, Italy

**Keywords:** cholesteatoma, biomarkers, endotyping, middle ear, chronic otitis media, precision medicine

## Abstract

Background: So far, no medical treatment is available for cholesteatoma (C) and the only effective therapy is complete surgical removal, but recurrence is common even after surgical treatment. While C is classically divided into two clinical phenotypes, congenital and acquired, only a few studies have focused on its potential biomarkers. This study aims to revise the literature to identify which biomarkers can define the endotype of C. Methods: We conducted a systematic review in accordance with the Preferred Reporting Items for Systematic Review and Meta-Analysis (PRISMA) process to identify published experimental articles about molecular biomarkers in C. Results: KGF and its receptor, MMP-9, KRT-1, KRT-10, and MIF might be considered biomarkers of recurrence, whereas Ki-67, TLR-4, RANKL, IL17, MMP-2, MMP-9, IL6, TNF-α, should be considered more specifically as biomarkers of bony erosion. Conclusions: These results are interesting especially from a prognostic point of view, nevertheless more studies are needed to search new biomarkers of C that could completely change not only the therapeutic standards of the disease, but also the clinical history of C itself in the era of precision medicine.

## 1. Introduction

C is a chronic middle ear (ME) disease, that can be defined as the accumulation of desquamating keratinized squamous epithelium within the ME accompanied by chronic inflammation [1]. C is considered a benign lesion with a gradual and destructive expansion. The proliferation of the epithelium and the erosion of the surrounding bone can lead to severe complications such as ossicular chain destruction, hearing loss, vestibular dysfunction, facial paralysis, and secondary infections as meningitis, brain abscesses, septic cavernous sinus thrombosis [2,3,4]. So far, no medical treatment is available for C and the only effective therapy is complete surgical removal, but recurrence is common even after surgical treatment. However, the reported relapse rate in the middle ear cholesteatoma (MEC) shows a wide range in the literature, from 0 to 72.4% and reflects patient’s age, surgical approaches, and minimum follow-up [5]. 

From a clinical point of view, C is a complex and heterogeneous disease often associated with chronic otitis media (COM) or some rare congenital syndromes affecting ear morphology and related pathologies [6]. For instance, the clinical expression of C may vary greatly. In terms of etiology, C has been mainly classified into two types: acquired and congenital C. Congenital cholesteatoma (CC) is described as a cystic mass with keratinizing squamous epithelium in the tympanic cavity or mastoid with an unbroken tympanic membrane (TM) and without a history of chronic otitis media (COM) or ME surgery [7]. CC is further classified into the closed-type, characterized by epithelial cysts without keratin exposure, and the open-type, characterized by ruptured cyst and proliferation of flat keratinizing epithelium. Whilst acquired C (AC) is classified into the *pars tensa* type and *pars flaccida* type based on its location [8]. AC is thought to be caused by the retraction of the TM, whereas CC seems to derive from an ectopic epidermoid at rest or ingrowth of meatal epidermis [9,10,11,12]. 

Furthermore, authors tried to differentiate subtypes/phenotypes of C basing on index of severity such as age of onset, uni- or bi-lateral disease, signs of local aggressiveness, and relapsing, missing a possible link with different pathophysiological mechanisms underlying the clinical manifestations because endotypes are not yet known. For the same reason, it is difficult to predict the clinical behavior of C, which may vary from a slow-progressing benign diseases to more aggressive disease [13]. Consequently, determining an appropriate follow-up period duration for patients with C is critical, and good prognostic biomarkers are needed in selecting patients who will require immediate surgical intervention [8].

Resuming literature data suggest that C is a heterogeneous condition in clinical presentations (phenotype) that may be related to several unknown underlying pathobiological mechanisms (endotype). In general, the phenotype defines any clinical observable features of a disease, without establishing a direct etiologic relationship with a distinct pathophysiologic mechanism, whereas the endotype describes different subgroups that share the same pathophysiologic processes leading to the development, the progression, and the presentation of a disease. Currently, only a few studies are available in the literature about biomarkers of C, some of which are mainly related to its etiopathogenesis and others to its clinical features as recurrence disease and bony erosion [2,14,15,16,17,18,19,20,21,22].

So far, no reviews have identified biomarkers of C endotyping. This study aims to revise the literature data in order to identify biomarkers defining the underlying pathobiology of the disease.

## 2. Materials and Methods

We conducted a systematic review in accordance with the PRISMA process [23] to identify published experimental and clinical articles about molecular biomarkers in C (Figure 1). Manuscripts were screened primarily by Ovid Medline and EMBASE and from other sources (Pubmed Central, Cochrane review, Web of Science, and Google Scholar) and published from January 2001 to June 2022. Literature searches were performed in July 2022. The authors focused on experimental studies matching the term as follow: [(cholesteatoma) OR (cholesteatomaous otitis media) OR (middle ear cholesteatoma) OR (chronic suppurative otitis media with cholesteatoma)] AND [(biomarker) OR (marker) OR (phenotype) OR (biochemistry) OR (signaling) OR (recurrence) OR (recidivism) OR (bone erosion) OR (bone destruction) OR (bone resorption) OR (complications)].

In the first screening, the authors read the title and abstract of articles selecting those being as inclusive as possible. The abstracts were screened independently by reviewers of the two groups. Inclusion and exclusion criteria were established before the selection of relevant studies. The inclusion criteria were primary research studies (including descriptive studies, observational studies, randomized trials, and basic science articles), published after January 2001. We excluded secondary research studies (e.g., review articles or systematic review), case studies, newspaper article, lecture, letter, comment, personal narrative, consensus conference, editorial. Only articles with full text available were included. Additional studies were manually identified from the reference lists of retrieved literature. We excluded all the articles that did not meet the inclusion criteria or deal directly with the issue investigated. We included only English-language peer-reviewed papers. 

## 3. Results and Discussion 

Our search yielded 2932 articles after duplicates removal. We excluded 1313 articles due to time of publication and type of article, and then 1619 were finally screened. This resulted in 9 publications which identified 13 molecular biomarkers. The full texts were assessed. We summarized in Table 1 the included studies. A biomarker has been defined as an objectively measured parameter that serves as an indicator of normal biological processes, pathogenic processes, or pharmacologic responses to a therapeutic intervention [24]. Recent literature has focused research on some biomarkers of C severity (in terms of recurrence and bone resorption). 

Literature on C showed some biomarkers associated with bone resorption and other biomarkers associated with C recurrence (Table 2 and Table 3). 

Most studies analyzed C specimens through immunohistochemistry and in situ hybridation, comparing results to normal retroauricular skin [14,17,18,20,21,22]. One study analyzed the expression of cytokeratins, TP53 and Matrix Metalloproteinasis-9 (MMP9) through quantitative real-time PCR [16]. Only one study evaluated serum level of MMP-2, MMP-9, and IL-6 in patients with COM [19]. 

Keratinocyte growth factor (KGF) is a member of fibroblast growth factor family involved in proliferation of epithelial cells and in wound healing. KGF and its receptor KGF-R play an important role in C pathogenesis [14]. Yamamoto-Fukuda investigated the possible involvement of KGF and KGF-R in the pathogenesis of C using in situ hybridization and immunohistochemistry, respectively. The results of this study indicated the possible involvement of both KGF and KGF-R in enhanced epithelial cell proliferative activity and recurrence of C [14].

In the same study, Yamamoto et al. evaluated the assessment of the proliferative activity of C using the labeling index for Ki-67. Ki-67 is a nuclear protein related to cellular proliferation and it is commonly used in mitotic index and tumor grading [15]. The authors showed a significantly higher Ki-67 labeling index in KGF/KGF-R cases than other cases [14]. According to Araz-Server et al., although the recurrent and non-recurrent C did not show significant differences in terms of the percentages of stained cells for either Ki-67 or pronuclear cell antigen (PCNA), the authors detected high Ki-67 staining in the malleus involvement group. They concluded that cell-proliferation markers could not be defined as indicators of recurrence of C, but they could be defined as indicators of destructive patterns of this disease [15].

Cytokeratins are proteins expressed in epithelial cells frequently used as biomarkers of cell migration, differentiation, and proliferation. Interactions between cytokeratins and other proteins can modulate signaling cascades involved in cell migration, invasion, and metastasis. It seems that their expression pattern correlates with the prognosis and metastatic potential of breast cancer and tumor of oral cavity [16]. Palko et al. investigated the expression of three cytokeratin genes (KRT1, KRT10 and KRT19), the matrix metalloproteinase 9 gene (MMP9) and the tumor suppressor TP53 gene in surgical samples of pediatric and adult C. They found that the expression of KRT1 and KRT10 was higher in pediatric recurrent C than in adults, whereas the expression of KRT19 was lower in more invasive recurrent C, both in pediatric and adult subjects. They also found significantly elevated expression of MMP9 in recurrent C in adults [16]. Toll-like receptors (TLR) are the first line mucosal defense, initiating the innate immune response their activation and subsequent inflammatory cytokine production play a key role in the osteoclast formation [17].

Si et al. [17] found a correlation between TLR4 expression in human acquired C and disease severity, invasion, bone destruction, and hearing loss [17]. Osteoclasts seem to modulate bone erosion in C. The differentiation and activation of osteoclasts are regulated by macrophage colony stimulating factor (M-CSF) and receptor activator of nuclear factor κB ligand (RANKL) [25]. Overproduction of RANKL is implicated in a variety of degenerative bone diseases, such as rheumatoid arthritis (RA) and periodontitis. In RA, osteoclasts are responsible for bone erosion and undergo differentiation and activation by RANKL [26]. In the study of Imai et al., the authors found a significantly larger number of osteoclasts in the eroded bone adjacent to C than in unaffected areas as well as peri-matrix fibroblasts expression of RANKL. They also showed increased concentration of interleukin (IL)-1β, IL6, tumor necrosis factor α (TNF α), and prostaglandin E2 in C. The presence of inflammatory cells in the C peri-matrix emphasizes the role of upregulation of factors related to osteoclast activation [18]. Accordingly, with these data in our experience increased expression (IL)-1β in specimens collected by patients affected by acquired C as compared to those with congenital C support the hypothesis of proinflammatory pathway activation in more symptomatic C (unpublished data). 

IL17, that has a role in pathogenesis of RA, induces the production of IL1, TNF-α and IL6 by synovial fibroblast, monocytes and macrophages and the up-regulation of matrix MMPO, nitric oxide and RANKL in chondrocytes and osteoblasts, leading to inflammatory bone destruction [27]. It seems that IL17 may induce bone destruction in patients with C through underlying mechanism of RA [20].

IL17 is also involved in type-3 immune response of CRS, directed against extracellular bacteria and fungi, where neutrophil activity may play a role in barrier damage [28].

Artono et al. [21] demonstrated a positive correlation between IL1a and TNF-a, and the severity of bone destruction in C patients. 

Macrophage migration inhibitory factor (MIF) serves as a regulator of innate and acquired immunities acting as a proinflammatory agent at sites of inflammation. MIF confers antimicrobial activity to macrophages against bacterial and viral infections. 

According to Choufani [22], MIF expression in human C is related to the presence of relapses. While previous research pointed out the role of different biomarkers in the C tissue, only Wu et al. analyzed the expression level of MMP-2, MMP-9, and IL6 in the serum of C patients [19]. This strategy may help to find easily accessible markers to use in clinical practice. MMP-9 is a protease involved in the degradation of extracellular matrix and basement membrane [29].

They found high expression levels of MMP-2, MMP-9 and IL-6 in the serum of C patients positively correlated with the injury degree of ossicle, which may be a sign of poor prognosis of C [19].

According to Wang et al. and in line with the *united airway* hypothesis, MMP-9 gene polymorphisms may influence susceptibility to the development of CRSwNP [30]. In addition, the expression levels of MMP-2, MMP-9 and IL-6 are closely related to the clinical manifestations and the severity of diabetic diseases [31]. In a more recent study, Yamamoto-Fukuda et al. also demonstrated that the use of an inhibitor of the complex Menin-MLL (MI-503) blocked the progression of middle ear C in vivo model [32].

Based on the recent literature on C biomarkers, KGF and its receptor, MMP-9, KRT-1, KRT-10, and MIF might be considered biomarkers of recidivism, whereas Ki-67, TLR-4, RANKL, IL17, MMP-2, MMP-9, IL6, TNF-α, and IL-1α might be biomarkers of bony erosion. MMP-9 could be a valid molecular biomarker of both recidivism and bony erosion. Furthermore, while MMP-9, KRT-1, KRT-10, MIF, Ki-67, TLR-4, RANKL, IL17, TNF-α, and IL-1α have been studied in tissue specimen, MMP-2, MMP-9, and IL6 have been found in serum. 

The expression of biomarkers may vary with age (adults vs. children), duration of the disease and type of sample. Unfortunately, we found only one paper in the literature analyzing biomarkers in adults vs. children [16]. Indeed, in the study by Palkò et al. the authors found increased expression of KRT1 and KRT10, in pediatric cases versus adults, especially in pediatric recurrent samples [16]. Conversely, the expression of KRT19 was lower in invasive recurrent cases in both the pediatric group and groups of adults. Furthermore, significantly elevated tissue MMP9 expression was found in recurrent cases in adults. This study demonstrated that the expression of cytokeratin distinguishes between pediatric/adult, non-recurrent/recurrent cases, suggesting that distinct differentiation state and cell division potential characterize them cases of C [16]. Likely, there is only one study in the literature measuring serum biomarkers, but no studies compare the expression of tissue biomarkers with those of serum [19].

Measuring serum markers is certainly desirable because it would make the study of endotyping more accessible and less invasive. 

However, future studies are needed to increase our knowledge of specific non-invasive diagnostic and prognostic tools. In addition, it would be very interesting to evaluate the level of biomarkers in different cohorts (adults/children), in different expressions of the C (recurrent/non-recurrent) and in different samples (tissue/serum) and, eventually, according to the duration of the C.

Furthermore, precise identification of specific biomarkers of aggressiveness could lead to a more personalized management (i.e., timing of intervention, recurrence prevention) and to a future identification of anti-growth/anti-proliferative agents as non-surgery therapeutic option.

## 4. Conclusions

Recent basic science research has led to significant advances in understanding the pathophysiological processes of disease. These processes define the endotype of a disease opening the way for the development of new diagnostic tools and innovative targeted treatments, especially in the rhinologic fields [33,34,35,36,37,38]. As has already happened for asthma and CRSwNP, and for many other autoimmune and oncological pathologies, the study of molecular biomarkers might help to develop new potential therapeutic target, and to improve an early diagnosis and a more accurate prognosis. In addition, molecular biomarkers might be used for a better scheduled follow-up.

Our data suggest that KGF and its receptor, MMP-9, KRT-1, KRT-10, and MIF might be considered biomarkers of recurrence, whereas Ki-67, TLR-4, RANKL, IL17, MMP-2, MMP-9, IL6, TNF-α, should be considered more specifically as biomarkers of bony erosion. These results are interesting, especially from a prognostic point of view; nevertheless, more studies are needed to search new biomarkers of C that could completely change not only the therapeutic standards of the disease, but also the clinical history of C itself.

## Figures and Tables

**Figure 1 jpm-12-01347-f001:**
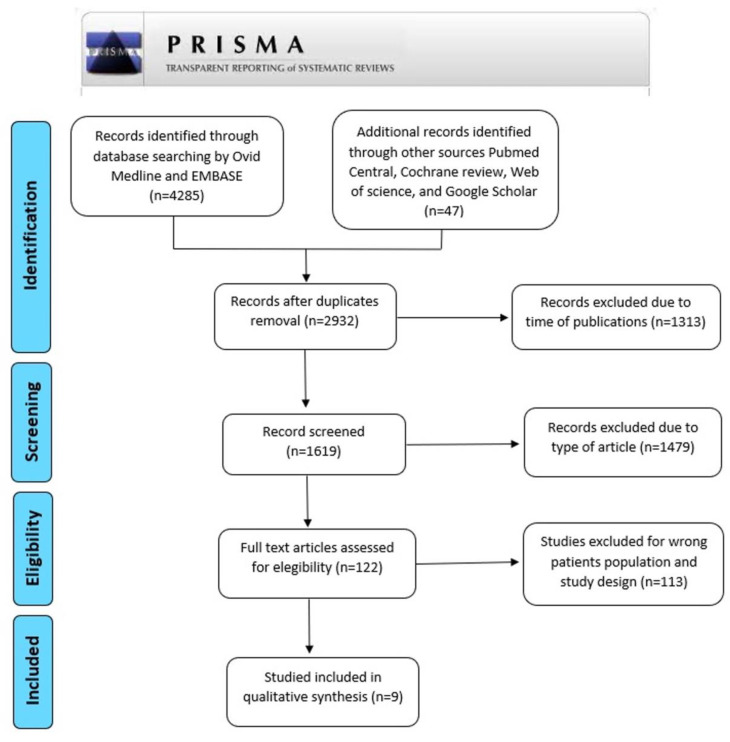
Prisma flow diagram of the systematic search.

**Table 1 jpm-12-01347-t001:** Studies on molecular biomarkers of Cholesteatoma.

Author Year Ref.	N. of Cases (Sex; Mean Age)	Methods	Sample Type	Analytical Technique	Relevant Results	Recurrence	Bone Erosion
**Yamamoto-Fukuda et al. 2003** [14]	A total of 56 cases (32 M, 24 F; mean age:35)	Expression of KGF, KGFR, Ki-67	Soft tissue	Immunohistochemical analysis and electroporatically transfection to the mice ears	KGF were detected in 80% of C, KGFR in 72%. Higher Ki-67 labeling index (66%) in KGF+/KGFR+ cases than other cases. Correlation between KGF+/KGFR+ expression and recurrence.	Y	
**Araz Server et al. 2019** [15]	A total of 43 cases (29 M, 14 F; mean age: 33)	Two groups: recurrent (21 patients) and non-recurrent within 2 years (22 patients). Ki-67 and the percentages of stained cells were calculated.	Bone tissue	Immunohistochemical analysis	A significant relationship between Ki-67 staining and malleus involvement		Y
**Palkó et al. 2018** [16]	A total of 26, 15 children and 11 adult (16 M, 11 F; mean age 23,4)	The expression of three cytokeratin genes (KRT1, KRT10 and KRT19), the matrix metalloproteinase 9 gene (MMP9) and the tumor suppressor TP53 gene was measured by qRT-PCR in surgical samples of cholesteatoma (C) cases and their expression level was compared to that of normal skin samples	Soft tissue	mRNA expression	Results showed identical expression pattern for KRT1 and KRT10, their expression was higher in pediatric cases than in adults, especially in pediatric recurrent samples. The expression level of KRT19 was lower in the more invasive recurrent cases both in our pediatric and adult groups. expression of MMP9 was highest in adult recurrent cases	Y	
**Si et al. 2015** [17]	A total of 187 specimens of acquired cholesteatoma (137 F and 51 M, mean age: 45)	Immunohistochemical analysis of TLR4 expression was performed in cholesteatoma specimen.	Congenital cholesteatomas and acquired cholesteatomas samples	Real-timePCR, Western blotting, and immunohistochemistry. Animal models	The number of TLR4-positive cells increased with an increased degree of cholesteatoma invasion, bone destruction, and hearing loss		Y
**Imai et al. 2019** [18]	A total of 24 cholesteatoma specimens from patients who underwent tympanomastoidectomy.	Osteoclasts were stained using tartrate-resistant acid phosphatase (TRAP)-hematoxylin counterstaining. The expression of RANKL mRNA was performed with the droplet digital polymerase chain reaction (ddPCR) system.	Bone-soft tissues	Immunohistochemistry, RNA Sequencing and ELISA	The number of osteoclasts on the bone surface adjacent to C was significantly larger than on the surface of control bone. The expression of RANKL mRNA was significantly higher in cholesteatomas than in control skin. RANKL was expressed in fibroblasts in the cholesteatoma perimatrix.		Y
**Wu et al. 2019** [19]	A total of 176 patients (91 M and 85 F, mean age 49, 65 ± 5, 27 with cholesteatomatous otitis media	The expression levels of MMP-2, MMP-9 and IL-6 in the serum of the selected patients were detected by ELISA	Serum	ELISA	MMP-2, MMP-9 and IL-6 were higly expressed in the serum of patients with C compared to the control group and were positively correlated with CT manifestations of the patients and the injury degree of ossicle, which may be a sign of poor prognosis of cholesteatomatous otitis media (COM).		Y
**Haruyama et al. 2010** [20]	Tissue specimens collected from 24 patients with cholesteatomas (16 M and 8 F, mean age: 44); congenital in 8 patients and acquired in 16 patients.	The expression and localization of IL17 and RANKL were examined by immunohistochemistry in tissue specimens. The cellular sources of IL-17 were assessed by double immunofluorescent staining with CD4. The level of IL-17 protein was determined using an enzyme-linked immunosorbent assay. The degree of bone destruction was compared with the IL-17 immunoreactivity	Cholesteatoma tissue	Immunohistochemistry	IL-17-positive inflammatory cells were seen in the subepithelial granulation tissue but not in the epithelium of the C. The localization of IL-17 expression coincided with CD4-positive lymphocytes. The subepithelial granulation had RANKL positive infiltrating cells and a significant correlation between IL-17- and RANKL-positive cells in the same specimens was recognized. The degree of bone destruction was dependent on the number of IL-17-positive cells that infiltrated the cholesteatoma.		Y
**Artono et al. 2019** [21]	A total of 46 patients (26 M and 20 F, age range of 21–30 years) with chronic suppurative otitis media (CSOM) with cholesteatoma	Pathological tissue in the form of cholesteatoma tissue and external acustic meatus skin during surgery was assessed. IL-1a expression was assessed by using ELISA. TNF-a expression was determined by immunohistochemical staining.	Cholesteatoma tissue	ELISA	There is a significant association between expression of TNF-a and IL-1a level on the severity of bone destruction in CSOM and C patients		Y
**Choufani et al. 2001** [22]	A total of 56 C specimens (33 M and 23 F), and 5 congenital and 51 acquired cases.	The immunohistochemical levels of expression of macrophage migration inhibitory factor (MIF) was statistically correlated to parameters of the other markers (galectin 1, -3, and -8, retinoid acid receptors (RAR) binding sites for sarcolectin, and invasion markers (cathepsins -B and -D, and matrix metalloproteinases [MMP]-2, -3, and -9)	Cholesteatoma tissue	Immunohistochemistry	MIF expression is higher in recurrent than in nonrecurrent cholesteatomas. MIF expression in infected ones it is correlated to MMP-3 and galectin-3 expression.	Y	

**Table 2 jpm-12-01347-t002:** Molecular biomarkers of recidivism.

Molecular Biomarkers	
KGF-KGFR	Yamamoto et al. 2003 [14]
MMP-9	Palkó et al. 2018 [16]
KRT-1	Palkó et al. 2018 [16]
KRT-10	Palkó et al. 2018 [16]
MIF	Choufani et al. 2001 [22]

**Table 3 jpm-12-01347-t003:** Molecular biomarkers of bone erosion.

Molecular Biomarkers	
Ki-67	Araz-Server et al. 2019 [15]
TLR-4	Si et al. 2015 [17]
RANKL	Imai et al. 2019 [18]
IL-17	Haruyama et al. 2010 [20]
MMP-2 (serum)	Wu et al. 2019 [19]
MMP-9 (serum)	Wu et al. 2019 [19]
IL-6 (serum)	Wu et al. 2019 [19]
TNF-α	Artono et al. 2019 [21]
IL-1α	Artono et al. 2019 [21]

## Data Availability

Data are available upon reasonable request.

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
