# Peer review of "Endotyping of Cholesteatoma: Which Molecular Biomarkers? A Systematic Review"

_jpm, 2022, doi:10.3390/jpm12081347_

Round 1

Reviewer 1 Report

The systemic review is focused on actual topic. The identification of molecular biomarkers could define endotypes of cholesteatomas with different expansion rate, bony erosion and risk of complications. The manuscript is well organized with comprehensive literature searches.

In the introduction I would recommend a minor revision. In the line 51 ...epithelium in the ME cavity or mastoid... The middle ear cavity consists of tympanic cavity, mastoid and Eustachian tube. I would recommend to use "tympanic cavity" instead of "ME cavity". In the lines 38 - 41 authors could add a citation of histological study of ossicular destruction by cholesteatoma Skoloudik, L.; Simakova, E; Kalfert, D; Chrobok, V.; Histological changes of the middle ear ossicles harvested during cholesteatoma surgery. Acta medica. 2015;58(4):119-122. 

I recommend to accept the manuscript for publication after minor revision.   

 identify which biomarkers can define the endotype of C. 

Author Response

Reviewer 1

I thank the reviewer for her/his precious advice.

In the introduction I would recommend a minor revision. In the line 51 ...epithelium in the ME cavity or mastoid... The middle ear cavity consists of tympanic cavity, mastoid and Eustachian tube.

Done

I would recommend to use "tympanic cavity" instead of "ME cavity". In the lines 38 - 41 authors could add a citation of histological study of ossicular destruction by cholesteatoma Skoloudik, L.; Simakova, E; Kalfert, D; Chrobok, V.; Histological changes of the middle ear ossicles harvested during cholesteatoma surgery. Acta medica. 2015;58(4):119-122. 

Added

Reviewer 2 Report

Thank you for inviting me to evaluate the article titled “Endotyping of Cholesteatoma: which molecular biomarkers?" The authors provided a systematic review of the literature evaluating biomarkers with the potential to define the endoype of cholesteatoma. The manuscript is well structured and the review follows the PRISMA guidlines. The topic itself has the potential to benefit the research community in this area as well as the readership. To be able to non-invasively sample and analyse tissue samples may open the possibility more precise diagnosis and targeted treatment of cholesteastoma.

However, the manuscript in its current format can still be improved by addressing the following minor suggestions.

1. Please consider to add the Prism Checklist as supplementary material (https://prisma-statement.org/PRISMAStatement/Checklist.aspx)

2. Row 43, MEC abbreviation is not explained

3. Table 1: For increased clarity, the authors may consider to add columns with information of sample type (soft tissue, bone+ soft tissue, serum, exudate etc) and analytical technique

4. What are the authors' considerations regarding the different biomarkers expression in terms of difference depending on age (adult vs. children), sample type, and duration of the cholesteatoma? Indeed, it is plausible that the biomarkers will be expressed differently depending on these and other factors. It would be of interest to discuss this and also in more depth discuss what studies are needed to increase the knowledge and the challenges to reach a situation where a non-invasive, specific diagnostic tool is available.

Author Response

 I thank you for comments to the systematic review giving me the possibility of resubmitting the revised manuscript

Please consider to add the Prism Checklist as supplementary material (https://prisma-statement.org/PRISMAStatement/Checklist.aspx)

Added

  1. Row 43, MEC abbreviation is not explained

Added

  1. Table 1: For increased clarity, the authors may consider to add columns with information of sample type (soft tissue, bone+ soft tissue, serum, exudate etc) and analytical technique

Added

  1. What are the authors' considerations regarding the different biomarkers expression in terms of difference depending on age (adult vs. children), sample type, and duration of the cholesteatoma? Indeed, it is plausible that the biomarkers will be expressed differently depending on these and other factors. It would be of interest to discuss this and also in more depth discuss what studies are needed to increase the knowledge and the challenges to reach a situation where a non-invasive, specific diagnostic tool is available.

thanks for your valuable suggestion. in this version we have added these important topics to the discussion as follows:

Biomarkers might de expressed differently based on age (adults vs children), duration of the C and type of sample analyzed. Unfortunately, we found only one paper in the literature analyzing biomarkers in adults vs children [16]. Indeed, in the study by Palkò et al. the authors found increased expression of KRT1 and KRT10, in pediatric cases versus adults, especially in pediatric recurrent samples [16]. Conversely, the expression of KRT19 was lower in invasive recurrent cases in both the pediatric group and groups of adults. Furthermore, significantly elevated tissue MMP9 expression was found in recurrent cases in adults. This study demonstrated that the expression of cytokeratin distinguishes between pediatric/adult, non-recurrent/recurrent cases, suggesting that distinct differentiation state and cell division potential characterize them cases of C [16]. Likely, there is only one study in the literature measuring serum biomarkers, but no studies compare the expression of tissue biomarkers with those of serum [19]. However, future studies are needed to increase our knowledge of specific non-invasive diagnostic and prognostic tools. In addition, it would be very interesting to evaluate the level of biomarkers in different cohorts (adults/children), in different expressions of the C (recurrent/non-recurrent) and in different samples (tissue/serum) and, eventually, according to the duration of the C.
